# Role of Cervical Elastography in Predicting Progression to Active Phase in Labor Induction in Term Nulliparous Women

**DOI:** 10.3390/diagnostics15040500

**Published:** 2025-02-19

**Authors:** Su-Jung Hong, Young-Mi Jung, Jeong-Eun Hwang, Ki-Su Lee, Geum-Joon Cho, Min-Jeong Oh

**Affiliations:** 1Department of Obstetrics and Gynecology, Korea University College of Medicine, Seoul 02841, Republic of Korea; crystalhsu1205@gmail.com (S.-J.H.); astroym@gmail.com (Y.-M.J.); md_cho@hanmail.net (G.-J.C.); 2Department of Obstetrics and Gynecology, Seoul National University Bundang Hospital, Seongnam 13620, Republic of Korea; 3Division of Medical Oncology, Department of Internal Medicine, College of Medicine, Korea University, Seoul 02841, Republic of Korea; hwangje02@sch.ac.kr; 4Department of Medical IT Engineering, College of Software Convergence, Soonchunhyang University, Soonchunhyango-ro, Asan-si 31538, Republic of Korea; 5Dongsan Medical Center, Keimyung University School of Medicine, Daegu 41931, Republic of Korea; cllgs315@naver.com

**Keywords:** labor induction, cervical elastography, active phase, sonoelastography, nulliparous pregnancy

## Abstract

**Background/Objectives:** Several factors, such as age, parity, body mass index, a favorable cervix, and fetal birth weight, are known to be related to the success of labor induction. With advancements in ultrasound technology, these factors have been studied to predict the success of vaginal delivery. However, there has been limited research on ultrasound measures that can effectively predict entry into the active phase of labor. Thus, we aimed to assess the use of cervical quantitative strain sonoelastography to predict entry into the active phase of labor induction. **Methods:** This prospective study included nulliparous term singleton pregnant women scheduled for labor induction between July 2018 and July 2022. Sonographic parameters were obtained using a transvaginal ultrasound approach with semiautomatic quantitative strain elastography software (E-Cervix; Samsung WS80A ultrasound device with a VR5-9 transducer, Samsung Medison Co., Ltd., Seoul, Republic of Korea), which provides objective measurements through the pixel-based analysis of elastographic maps. Univariate and multivariate logistic regression and area-under-the-curve analyses were used to evaluate the diagnostic performance of the variables under consideration in predicting the onset of the active phase of labor. **Results:** A total of 71 women were included in the study, and 29 progressed to the active phase. The cervical length, angle of progression, and mean strain from the external cervical os were significantly associated with successful entry into the active phase. The receiver operating characteristic (ROC) curve model also indicated a higher predictive value when the elastographic parameters were combined. **Conclusions:** Cervical elastography can be used as a sonographic index to predict progression to the active phase of labor.

## 1. Introduction

The induction of labor (IOL) is the stimulation of contractions before the spontaneous onset of labor and is one of the most common procedures in obstetrics, with rates varying significantly across countries, ranging from 20% to 35% of deliveries according to previous data [1,2,3]. Labor induction rates in the United States have increased significantly from 9.6% in 1990 to 41% in 2022, reflecting a continued upward trend in obstetric interventions [4,5]. The decision to induce labor is typically made when it is deemed that the advantages for either the mother or the fetus outweigh the benefits of continuing pregnancy. Common reasons for labor induction include the premature rupture of membranes (PROM), gestational hypertension, oligohydramnios, non-reassuring fetal status, post-term pregnancy, and various maternal medical conditions [6].

Various factors play significant roles in determining the success of labor induction in achieving vaginal delivery. Positive factors that enhance the likelihood of success include younger age, having previous childbirth experience, having a lower body mass index (BMI), possessing a favorable cervix, and having a baby with a birth weight of <3500 g [7,8]. However, the efficacy of labor induction is often influenced by the duration of the induction process, particularly when the cervix is unfavorable [9].

For these reasons, some women fail to enter the active phase, even with the infusion of oxytocin, and end up undergoing a cesarean section. Given that various patient characteristics and pelvic anatomies can influence the success of labor induction, numerous studies have been conducted to predict the likelihood of successful vaginal delivery [10].

To date, the Bishop score has been the traditional method used to evaluate cervical status; however, it is subjective and causes discomfort to patients [11], and its primary limitation lies in its poor reproducibility and significant inter-examiner variability [12]. Recent research has focused on incorporating the Bishop score into ultrasound frameworks, including the development of E-Cervix software, to achieve more objective measurements [13]. Despite these advances, there remains a need for more objective methods to predict cervical characteristics. Consequently, several studies have attempted to use ultrasound to assess cervical condition, and previous research has indicated that a longer cervical length is associated with a lower likelihood of successful labor induction [14].

Along with cervical length, several elastography tools have been developed for the objective assessment of cervical condition. Two main elastography techniques are currently used: strain elastography and shear wave elastography (SWE). While strain elastography measures tissue deformation in response to external compression and provides qualitative or semi-quantitative assessments, SWE uses acoustic radiation force to generate shear waves and provides quantitative measurements of tissue stiffness in kilopascals (kPa) [15]. SWE is an ultrasound-based method that allows for the direct estimation of cervical tissue density without the need for manual compression [16,17]. Research has shown that cervical tissue stiffness tends to decrease with gestational age [18]. Thus, our study aimed to explore the use of ultrasonography and elastography to predict entry into the active phase during labor induction.

## 2. Methods

### 2.1. Study Population

This prospective study included nulliparous women scheduled for labor induction between July 2018 and July 2022 at the Korea University Guro Hospital. Gestational age was determined using ultrasonography during the first trimester. The inclusion criteria were as follows: (1) singleton pregnant women aged 19–45 years; (2) gestational age ≥ 37 weeks; (3) women with no contraindications for IOL; and (4) women with an unfavorable cervix (Bishop score < 7). Women with a history of cervical insufficiency or cervical surgery and women with fetal malformations were excluded. This study was approved by the Institutional Review Board of the Korea University Medical Center (IRB No. 2018GR0236, IRB date of approval: 25 July 2018).

### 2.2. Labor Induction Protocol

Labor induction was performed according to our institutional protocol. Prior to induction, all patients underwent cervical assessment using the Bishop score. For patients with an unfavorable cervix (Bishop score ≤ 6), cervical ripening was initiated using prostaglandin E2 (dinoprostone vaginal insert, 10 mg). Oxytocin infusion was initiated either after cervical ripening or directly in patients with a favorable cervix (Bishop score > 6). Oxytocin was administered using a standardized protocol starting at 2 mU/min, with increments of 2 mU/min every 30 min until adequate uterine contractions were achieved (3–5 contractions per 10 min) or a maximum dose of 20 mU/min was reached. The artificial rupture of membranes (amniotomy) was sometimes performed when the cervix was favorable (≥6 cm dilated) and the fetal head was engaged to the cervix, unless spontaneous rupture had already occurred. Throughout the induction process, continuous fetal heart rate monitoring was maintained, and the progress of labor was assessed by cervical examination every hour or as clinically indicated. For patients who failed to enter active labor after the initial induction, the same induction protocol was repeated 24 h later.

### 2.3. Variables and Outcomes

Maternal characteristics and obstetric history data were obtained from the Korea University Medical Center (KUMC). The maternal height and weight were measured on the day of admission. Clinical obstetricians evaluated the Bishop score and sonographic parameters within 1 week before admission. Neonatal and delivery outcomes were also extracted from the medical records of KUMC. The primary outcome was entering the active phase, defined as cervical dilatation ≥6 cm with regular contractions within 12 h from the infusion of oxytocin. We also evaluated the success of vaginal delivery in mothers who did not enter the active phase within 12 h but continued with labor afterward.

### 2.4. Ultrasound Measurement and Cervical Elastography

The cervix of the participants was evaluated before admission for IOL. In addition to cervical length, we measured the posterior cervical angle, which is defined as the angle between the cervical canal and the posterior uterine wall. The elasticity contrast index (ECI), hardness ratio (HR), mean strain from the internal os (IOS) and external os (EOS), and IOS/EOS ratio were obtained using a transvaginal ultrasound approach with semiautomatic software (E-Cervix; Samsung Medison Co., Ltd., Seoul, Republic of Korea). Detailed descriptions of each parameter are provided in a separate paper [19]. For each participant, a cervical image was obtained three times, and the average value was calculated. The scan was performed by a trained obstetrician sonographer with more than 10 years of experience. Our standardized protocol for E-cervix elastography measurements included several key steps to ensure optimal image acquisition and quality:

1.Pre-examination Preparation

Patients were instructed to void their bladder before the examination to optimize image quality. All examinations were performed with the patient in the lithotomy position.

2.Image Acquisition Technique

The ultrasound probe was positioned to display the cervical apex at the monitor’s superior aspect, with fetal components orientated to the left of the image sector. Following Fetal Medicine Foundation guidelines, we obtained cervical images using the same plane as standard cervical length measurement, taking care to minimize the anterior cervical pressure.

3.Quality Assurance

Image quality was monitored through the system’s reliability indicators. The probe was maintained in a stable position until all motion indicators displayed acceptable status (green indicators). Patients were instructed to maintain normal breathing patterns during image acquisition. Images affected by fetal movement, particularly in cases of breech presentation, were discarded and repeated to ensure measurement accuracy.

4.Region of interest (ROI) Placement Protocol

ROI markers were positioned using a grayscale reference image for optimal precision.

The endocervical canal was delineated using either a two-point or four-point ROI system, depending on cervical curvature. The measurement area was carefully defined to encompass the entire cervical tissue while excluding adjacent structures such as bladder tissue or the vaginal wall.

### 2.5. Statistical Analyses

Data were reported as numbers for categorical variables and medians for continuous variables. Numerical variables, including maternal age, BMI, Bishop score, CL, estimated fetal weight, posterior cervical angle, angle of progression (AOP), ECI, HR, IOS, EOS, gestational age, and birth weight, were compared using Welch’s two-sample *t*-test. Two-proportion Z-tests were performed for equal proportions to compare the rates of dinoprostone use, vaginal delivery, and neonatal sex. Statistical significance was set at *p* < 0.05. Generalized linear models were built to assess the odds ratios (ORs) of variables for outcomes, including vaginal delivery and entrance to the active phase. ORs, their 95% confidence intervals, and *p*-values were calculated with or without adjustments for maternal age, maternal BMI at gestation, gestational age, and dinoprostone use. Variables were log-normalized where appropriate. Multivariate logistic regression and area-under-the-curve analyses were used to evaluate the diagnostic performance of the variables under consideration in predicting the onset of the active phase of labor. R statistics software version 4.3.1 (R Foundation for Statistical Computing, Vienna, Austria) was used for the statistical analysis.

## 3. Results

### 3.1. Study Population

After excluding ineligible patients, 71 women were included in the study. The indications for labor induction in our study population included the premature rupture of membranes (PROM) (n = 20, 28.2%), low-risk nulliparous women after 39 weeks of gestation (n = 27, 38.0%), gestational diabetes mellitus (GDM) (n = 6, 8.5%), hypertensive disorders of pregnancy (HDPs) (n = 5, 7.0%), intrauterine growth restriction (IUGR) (n = 6, 8.5%), large for gestational age (LGA) (n = 2, 2.8%), and post-term pregnancy (n = 5, 7.0%). Among them, 29 (40.8%) reached the active phase, while 42 (59.2%) failed to enter the active phase. In the group that entered the active phase, 86.2% successfully achieved a vaginal delivery. The median time to the active phase was 10 h (IQR 8–14). There were no statistically significant differences in maternal age or pre-pregnancy BMI between the groups that entered the active phase and those that did not. However, at the time of delivery, the BMI was lower in the group that entered the active phase (27.9% vs. 26.1%, *p* < 0.05), and there were differences in the Bishop score, as well. The use of dinoprostone was more common in the group that did not enter the active phase (69.0% vs. 41.4%, *p* < 0.05) (Table 1).

As shown in Table 1, when examining the ultrasonographic findings obtained during induction, it was observed that the group that entered the active phase had a shorter CL (2.9 cm vs. 2.2 cm, *p* < 0.001), which could be anticipated. The estimated fetal weight did not differ between the two groups; however, the AOP was greater in the group that entered the active phase (96° vs. 103°, *p* < 0.01). Although there were no differences in the other elastography parameters, the mean strain from external os (EOS) was higher in the group that entered the active phase (0.276 vs. 0.330, *p* < 0.05).

Regarding delivery outcomes, the group that entered the active phase had a smaller birthweight (3.42 kg vs. 3.25 kg, *p* < 0.05), while there were no differences in sex distribution or the number of deliveries.

### 3.2. Predictors of Successful Entry into the Active Phase

Table 2 presents the univariate and multivariate analyses of sonographic factors that could influence entry into the active phase. In the univariate analysis, the AOP (OR, 1.07; 95% CI = 1.02–1.12), CL (OR, 0.294; 95% CI = 0.134–0.575), EOS (OR, 1.68; 95% CI = 1.03–2.90), and Bishop score (OR, 2.76; 95% CI = 1.62–5.36) showed a significant association with entering the active phase. However, other elastographic measures, such as ECI, HR, IOS/EOS ratio, and IOS, were not associated with entry into the active phase. Table 3 presents the univariate and multivariate analyses of the factors related to the success of induction. Regarding the success of induction, statistically significant correlations were observed with AOP (OR, 1.074; 95% CI = 1.025–1.134), ECI (OR, 1.832; 95% CI = 1.060–3.352), CL (OR, 0.517; 95% CI = 0.274–0.914), EOS (OR, 1.96; 95% CI = 1.16–3.56), and Bishop score (OR, 2.56; 95% CI = 1.54–4.66). Among the ultrasound variables, we used the IOS/EOS ratio as a representative variable for IOS and EOS to avoid multicollinearity when building multivariate models because the three variables were closely related between one another.

Figure 1 shows the area under the curve (AUC) of each model to predict successful labor induction. The AUC of Model 1, which included maternal age, BMI, and gestational age, was 0.720, with a sensitivity of 55.2% and specificity of 83.3% (Appendix A). The AUC of Model 2, which combines Model 1 and the Bishop score (≥3), was 0.808 with a sensitivity of 55.2% and specificity of 85.7%. The AUC of Model 3, which is Model 2 and the IOS/EOS ration (≥1), shows a sensitivity of 69% and specificity of 85.7%. The AUC increased from 0.720 in Model 1 to 0.834 in Model 3, and the Delong test showed a significant difference in the *p*-value.

## 4. Discussion

The major findings of this study were as follows: (1) those who entered the active phase within 12 h of oxytocin infusion tended to have higher Bishop scores and a lower BMI at delivery. In terms of sonographic parameters, the CL was shorter when the AOP and EOS values were higher. In the case of delivery variables, vaginal delivery was more successful, and the birth weight of the neonates was lower. (2) In the univariate analysis, the factors that affected active phase entry were the AOP, CL, EOS, and Bishop score. (3) In the linear regression model with the Bishop score and IOS/EOS ratio, the performance in terms of the AUROC (area under the receiver operating characteristic) was 0.834, which was higher than that for maternal characteristics only. The performance of the AUROC was significantly valuable.

Cervical tissue changes by softening, dilating, and effacement as the onset of labor approaches [11]. Traditionally, cervical examination using the Bishop score is the only method used to evaluate cervical status. However, it has been shown to be subjective and demonstrates a relatively low predictive performance [10]. Therefore, our study used elastography, an objective method of assessing the relative consistency of tissues that allows the visualization of stiffness by color coding and enables the comparison of different parts of tissues [20].

Our findings demonstrate how sonographic and elastographic parameters can enhance the prediction of entry into the active phase and the success of IOL. In addition to the angle of progression [21], elastographic parameters such as the EOS and ECI show promise as predictive tools for vaginal delivery.

Previous studies have shown that elastography can be a useful tool in predicting the onset of labor induction or the success of vaginal delivery. Lu et al. [10]. concluded that the combination of sonographic CL and SWE was superior to the Bishop score in predicting IOL failure. Recent advances have focused on integrating ultrasound parameters into traditional assessment methods to create more objective evaluation systems. From this point of view, our study enhanced previous findings in different ways. First, we created models to illustrate how the AUC changed when the sonographic and elastographic parameters were added. Second, the DeLong test was used to evaluate the significance of each model.

E-cervix (Samsung Medison Co., Ltd.) is a semi-automatic software that analyzes the strain ratio between the internal and external orifices of the cervix using vibrations caused by natural internal movements. The advantage of this technique is that the generation of the mechanical impulse is operator-independent, which improves reproducibility and reduces interobserver variability [22]. A comprehensive literature review of E-Cervix-related research revealed several key developments in this field. Early studies focused on validating the E-Cervix module as an objective tool for cervical assessment, demonstrating its potential to overcome the limitations of traditional Bishop score evaluation [19,23,24,25]. Several studies have explored the predictive value of E-Cervix measurements for various obstetric outcomes. Particularly noteworthy is the emerging evidence regarding the hardness ratio (HR) as a predictive marker. Nazzaro et al. (2024) demonstrated that women with a low HR, especially those with values less than 50% or 35%, showed an increased risk of preterm birth (PTB). Their findings revealed that women who delivered preterm had a significantly higher HR compared to those who carried to term, along with a notably lower internal os strain (IOS) and external os strain (EOS) [17,26]. The predictive value of elastography extends beyond just preterm birth prediction. Rizzo et al. found that HR assessment through sonoelastography enhanced the predictive accuracy of cervical length measurements for imminent delivery in nulliparous women approaching term. This suggests that combining traditional cervical length measurements with elastography parameters might provide a more comprehensive risk assessment [22]. Furthermore, He et al. emphasized that in singleton pregnancies with a short cervix receiving progesterone therapy, cervical length measurement alone may be insufficient for the accurate risk assessment of spontaneous preterm birth [27]. Their research highlighted the importance of evaluating cervical stiffness, particularly focusing on internal and external os strain measurements. In specific high-risk populations, such as women with a history of Loop Electrosurgical Excision Procedure (LEEP), Cha et al. demonstrated that cervical strain measurement in mid-trimester could be particularly valuable. They found that previous LEEP procedures were associated with changes in cervical strain and cervical length shortening, establishing elastography as a useful tool for predicting sPTB in this population [28]. Furthermore, others have suggested its ability to predict spontaneous preterm delivery [29], the success of IOL [11], and failure to enter the active phase [10]. This module combines various ultrasound parameters to provide a more reproducible and objective evaluation of cervical status, addressing the limitations of the traditional Bishop score. Efforts to translate subjective clinical parameters into quantifiable ultrasound measurements have shown promising results in predicting induction outcomes. Based on prior research, we compared the predictive performance curves based on maternal baseline characteristics with sonographic and elastographic parameters. Our analysis demonstrated higher performance values for elastographic parameters.

Unnecessary labor pain experienced by patients is a concern for everyone. Predicting the onset of the active phase and identifying patient outcomes are crucial clinical issues. However, in previous studies, the predictions were not accurate and relied solely on the Bishop score or CL. This study aimed to enhance the predictive performance of the onset of the active phase by incorporating a newly developed technology called elastography, along with clinical factors and existing ultrasound variables. This improvement could potentially offer clinical assistance. Further research involving a larger sample size and the minimization of differences between the two groups is required to refine this model. Another point to consider is that our findings should be interpreted in the context of recent research showing that the risk of intrapartum fetal compromise requiring cesarean delivery varies significantly based on the indication for induction, with particularly high rates observed in cases of fetal growth restriction (17.2%) and combined preeclampsia and fetal growth restriction (23.4%). These findings, together with the FMF preeclampsia risk assessment, can help us to better counsel patients on the likelihood of successful vaginal delivery following the induction of labor [30]. In our study population of 71 cases, we observed 4 cases (5.6%) requiring cesarean delivery due to fetal distress, distributed across different indications for induction, highlighting the importance of careful monitoring during labor induction regardless of the initial indication.

The main strength of our study was that only one sonographer professionally checked all the sonographic parameters. Thus, we did not need to consider intraobserver or interobserver reproducibility. Additionally, there are advantages to studies that collect data prospectively. However, this study had some limitations. First, this was a single-center study, which may not represent the entire population. Second, the sample size was not sufficiently large to represent the population; thus, more studies with larger populations are required. Third, a key limitation concerns the reproducibility of E-Cervix measurements, particularly the EOS parameter. As demonstrated by Mlodawski et al., while most E-Cervix parameters show good reproducibility, EOS measurements demonstrate relatively lower reproducibility compared to other parameters [13]. This variability in EOS measurements could potentially affect the reliability of the IOS/EOS ratio used in our multivariable model. To minimize this limitation in our study, all measurements were performed by experienced sonographers who underwent standardized training, and multiple measurements were taken for each parameter. However, future studies should consider this inherent variability when interpreting EOS-related measurements and perhaps explore alternative parameters or measurement techniques that might offer better reproducibility.

## 5. Conclusions

In conclusion, our study demonstrates that cervical elastography, particularly when combined with traditional clinical parameters and sonographic measurements, can effectively predict entry into the active phase of labor induction. The combination of maternal characteristics, Bishop score, and elastographic parameters (IOS/EOS ratio) showed a superior predictive performance (AUC 0.834) compared to maternal characteristics alone. While our study was limited by its single-center design and sample size, the use of a single experienced sonographer and prospective data collection strengthened our findings. The implementation of E-cervix, a semi-automatic software that reduces operator dependency, suggests that elastographic assessment could be a valuable tool in clinical practice for predicting labor induction outcomes. Future studies with larger populations across multiple centers are needed to validate these findings and potentially improve the management of labor induction.

## Figures and Tables

**Figure 1 diagnostics-15-00500-f001:**
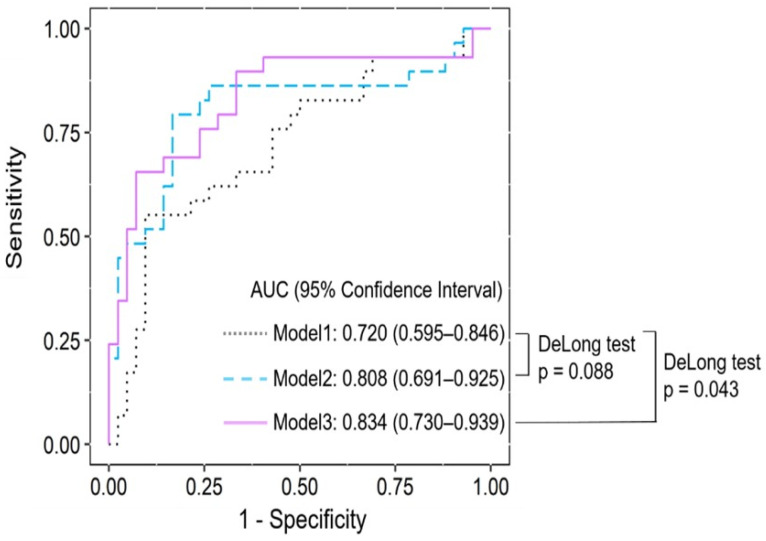
The linear regression models’ prediction performance in terms of AUROC for successful induction. Model 1, maternal age, maternal BMI, and gestational age; Model 2, Model 1 and Bishop score (≥3); Model 3, Model 2 and IOS/EOS ratio (≥1).

**Table 1 diagnostics-15-00500-t001:** Baseline characteristics and delivery outcomes of study population.

	Failed to Enter the Active Phase (N = 42)	Entered the Active Phase (N = 29)	*p*-Value
**Maternal baseline characteristics**			
Maternal age (years)	34 (31–36)	33 (31–36)	0.702
BMI before pregnancy (kg/m^2^)	22.9 (21.0–25.6)	21.6 (19.6–23.3)	0.087
BMI at delivery (kg/m^2^)	27.9 (26.6–33.2)	26.1 (24.3–29.7)	0.013 *
Dinoprostone use	29 (69.0%)	12 (41.4%)	0.038 *
Bishop score	2 (1–2)	2 (2–3)	<0.001 *
**Ultrasound variables**			
Cervical length (cm)	2.94 (2.20–3.47)	2.20 (1.59–2.49)	<0.001 *
Estimated fetal weight (kg)	3.38 (3.19–3.65)	3.33 (2.95–3.69)	0.386
PCA (degree)	111 (98–130)	125 (110–139)	0.083
AOP (degree)	96 (90–103)	103 (96–112)	0.006 *
ECI	3.74 (3.01–4.41)	3.82 (3.25–4.70)	0.207
HR	58.0 (43.5–71.7)	55.5 (42.8–63.8)	0.242
IOS	0.365 (0.278–0.477)	0.355 (0.287–0.427)	0.686
EOS	0.276 (0.235–0.332)	0.330 (0.280–0.397)	0.047 *
**Delivery variables**			
Gestational age (days)	278 (272–281)	281 (277–284)	0.073
Vaginal delivery	19 (45.2%)	25 (86.2%)	<0.001 *
Birth weight (kg)	3.42 (3.17–3.64)	3.25 (2.89–3.55)	0.022 *
Neonatal sex (male)	19 (45.2%)	15 (51.7%)	0.591

Data are presented as medians (interquartile ranges) and numbers (percentages) where applicable. Abbreviations: AOP, angle of progression; BMI, body mass index; ECI, elasticity contrast index; EOS, mean strain from external os; HR, hardness ratio; IOS, mean strain from internal os; PCA, posterior cervical angle. *p*-values < 0.05 are shown with an asterisk.

**Table 2 diagnostics-15-00500-t002:** Univariate and multivariate analysis of elastographic and sonographic parameters to predict entry into the active phase.

Variables	Univariate OR and 95% CI	*p*-Value	Multivariate OR and 95% CI	*p*-Value ^§^
PCA	1.02 (0.997–1.04)	0.099	1.02 (0.998–1.05)	0.080
AOP	1.07 (1.02–1.12)	0.009 *	1.05 (0.996–1.11)	0.084
ECI	1.41 (0.841–2.42)	0.199	1.31 (0.727–2.42)	0.372
HR	0.982 (0.951–1.01)	0.257	0.986 (0.948–1.02)	0.446
IOS/EOS ratio	0.317 (0.074–1.11)	0.092	0.361 (0.071–1.50)	0.184
Cervical length	0.294 (0.134–0.575)	0.001 *	0.302 (0.126–0.633)	0.003 *
IOS (normal) *	0.994 (0.614–1.61)	0.979	0.916 (0.507–1.63)	0.765
EOS (normal) *	1.68 (1.03–2.90)	0.048 *	1.51 (0.873–2.74)	0.152
Bishop score	2.76 (1.62–5.36)	<0.001 *	2.66 (1.44–5.54)	0.004 *

IOS and EOS were log-normalized: IOS (normal) = (log(IOS) − mean(log(IOS)))/stdev(IOS). ^§^ Adjusted for maternal age at gestation, maternal body mass index at gestation, gestational age, and dinoprostone usage. Abbreviations: AOP, angle of progression; ECI, elasticity contrast index; EOS, mean strain from external os; HR, hardness ratio; IOS, mean strain from internal os; OR, odds ratio; PCA, posterior cervical angle. *p*-values < 0.05 are shown with an asterisk.

**Table 3 diagnostics-15-00500-t003:** Univariate and multivariate analysis of elastographic and sonographic parameters to predict the success of the induction of labor.

Variables	Univariate OR and 95% CI	*p*-Value	Multivariate OR and 95% CI	*p*-Value ^§^
PCA	1.006 (0.985–1.029)	0.581	1.01 (0.985–1.03)	0.480
AOP	1.074 (1.025–1.134)	0.005 *	1.06 (1.01–1.13)	0.033 *
ECI	1.832 (1.060–3.352)	0.037 *	1.99 (1.07–3.99)	0.038 *
HR	0.982 (0.949–1.013)	0.259	0.982 (0.946–1.02)	0.314
IOS/EOS ratio	0.525 (0.152–1.71)	0.288	0.403 (0.099–1.49)	0.180
Cervical length	0.517 (0.274–0.914)	0.030 *	0.628 (0.314–1.19)	0.166
IOS (normal) *	1.32 (0.812–2.19)	0.269	1.21 (0.696–2.12)	0.503
EOS (normal) *	1.96 (1.16–3.56)	0.017 *	2.14 (1.20–4.08)	0.014 *
Bishop score	2.56 (1.54–4.66)	<0.001 *	2.78 (1.52–5.67)	0.002 *

IOS and EOS were log-normalized: IOS (normal) = (log(IOS) − mean(log(IOS)))/stdev(IOS). ^§^ Adjusted for maternal age at gestation, maternal BMI at gestation, gestational age, and dinoprostone usage. Abbreviations: AOP, angle of progression; BMI, body mass index; ECI, elasticity contrast index; EOS, mean strain from external os; HR, hardness ratio; IOS, mean strain from internal os; OR, odds ratio; PCA, posterior cervical angle. *p*-values < 0.05 are shown with an asterisk.

## Data Availability

The authors declare that the data obtained in this research are available from the corresponding authors upon reasonable request.

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
