# Peer review of "Role of Cervical Elastography in Predicting Progression to Active Phase in Labor Induction in Term Nulliparous Women"

_diagnostics, 2025, doi:10.3390/diagnostics15040500_

Round 1
Reviewer 1 Report
Comments and Suggestions for Authors
I was presented with a manuscript for review titled: “Role of Cervical Elastography in Predicting Progression to Active Phase in Labor Induction in Term Nulliparous Women.”
Below, I outline my comments on the submitted article:
The abstract is well-written, though I have a few suggestions. First, when discussing the E-Cervix software, it would be beneficial to specify the type of elastography used. This clarification is important because readers may not be familiar with the elastography type represented by E-Cervix. In this case, we are dealing with quantitative strain elastography, which is a crucial point since strain elastography has traditionally been presented in a semi-quantitative scale model. However, E-Cervix employs software capable of objectively providing quantitative data based on pixel counts in the elastographic map—an approach previously exclusive to shear wave elastography. I suggest including a statement that this is quantitative strain elastography.
Additionally, there is a missing parenthesis in the sentence:
“The Receiver Operating Characteristic (ROC) curve model also indicated a higher predictive value” – ensure the parenthesis is closed.
1. Introduction
“...comprising more than 20% of vaginal deliveries[1].” – The percentage varies significantly depending on the country studied, as noted in WHO data. I recommend adding this context.
“Rates in the United States have increased significantly from 9.6% in 1990 to 25.7% in 2018.” – It would be worthwhile to reference more recent data than 2018, as these figures are updated annually in U.S. statistical reports.
Phrases such as “ruptured membranes without labor” and “pregnancies that have gone past their due date” should use standardized medical terminology, such as Postterm pregnancy or PROM (Premature Rupture of Membranes).
“The Bishop score has been the sole method used to evaluate cervical status; however, it is subjective and causes discomfort to patients.” – In my opinion, the main limitation of the Bishop score is its lack of reproducibility in quantitative terms and the significant variability between examiners (see Faltin-Traub, E.F.; Boulvain, M.; Faltin, D.L.; Extermann, P.; Irion, O. Reliability of the Bishop score before labour induction at term. Eur. J. Obstet. Gynecol. Reprod. Biol. 2004, 112, 178–181). There is ongoing research aimed at objectifying this scale by incorporating it into ultrasound frameworks, including the use of E-Cervix software (DOI: 10.3390/jcm12134492). I believe this is a point worth addressing in the introduction.
In the introduction, the authors also discuss SWE (shear wave elastography). It would be helpful to differentiate between SWE and strain elastography, as this distinction is key to understanding the E-Cervix module.
Methods
In this section, the authors should first describe the protocol for labor induction. Specifically, detail the criteria for oxytocin infusion initiation (e.g., based on the Bishop score), whether preinduction methods were used (e.g., prostaglandins, laminaria, Foley catheter), and under what circumstances amniotomy was performed.
“The detailed descriptions of each parameter are provided in a separate paper.” – A reference to the appropriate publication must be provided. For instance, the parameters are described here: (DOI: 10.1038/s41598-021-02498-3).
The methodology should also include a more detailed description of the imaging technique used for E-Cervix evaluation. In this context, the pressure applied by the probe to the cervix is critically important and should be specified.
Results
“Among them, 29 reached the active phase, while 42 failed to enter the active phase.” – Percentages should be provided here.
The term posterior cervical angle appears in the tables for the first time but is not mentioned earlier in the text. This term should be introduced and explained earlier in the manuscript.
Given the low reproducibility of the Bishop score between observers, it would be helpful to present it as a categorical variable in the results. For instance, literature commonly uses a cutoff of BS <7 and ≥7 to define an unfavorable or favorable cervix, respectively.
Discussion
“Previous studies have shown that elastography can be a useful tool for predicting the onset of labor induction or the success of vaginal delivery.” – As mentioned earlier, attempts have been made to reflect the Bishop score using ultrasound parameters and the E-Cervix module. This effort to objectify the evaluation before induction of labor (IOL) should be mentioned in the discussion (DOI: 10.3390/jcm12134492).
The authors should provide a broader literature context for their study. For example, a Medline search using the keyword "E-Cervix" yields 27 results.
Additionally, the discussion should address the limitations of the study. In my opinion, one limitation might be the reproducibility of E-Cervix results. While the parameters generally show good reproducibility, the poorest reproducibility is observed for EOS, which affects the reliability of the IOS/EOS parameter included in the multivariable model (DOI: 10.3390/jcm12134492).
General Comments
Overall, the study is good and relatively innovative, as there is limited literature on this method. However, it requires some improvements.
Author Response
Response to Reviewer:
Reviewers' Comments 1
#Point 1. : Below, I outline my comments on the submitted article: The abstract is well-written, though I have a few suggestions. First, when discussing the E-Cervix software, it would be beneficial to specify the type of elastography used. This clarification is important because readers may not be familiar with the elastography type represented by E-Cervix. In this case, we are dealing with quantitative strain elastography, which is a crucial point since strain elastography has traditionally been presented in a semi-quantitative scale model. However, E-Cervix employs software capable of objectively providing quantitative data based on pixel counts in the elastographic map—an approach previously exclusive to shear wave elastography. I suggest including a statement that this is quantitative strain elastography. Additionally, there is a missing parenthesis in the sentence:
“The Receiver Operating Characteristic (ROC) curve model also indicated a higher predictive value” – ensure the parenthesis is closed.
Response: Thank you for your valuable feedback. We appreciate your thorough review and have addressed your suggestions as follows:
We will clarify that E-Cervix utilizes quantitative strain elastography. The revised text will specify that while traditional strain elastography provides semi-quantitative measurements, E-Cervix employs advanced software that generates objective quantitative data through pixel count analysis of the elastographic map - a capability previously associated only with shear wave elastography.
The corrected sentence now reads:
“Thus, we aimed to determine the use of cervical quantitative strain elastography to predict entry into the active phase of labor induction.”
“Sonographic parameters were obtained using a transvaginal ultrasound approach with semiautomatic quantitative strain elastography software (E-Cervix; Samsung Medison Co., Ltd., Seoul, Korea), which provides objective measurements through pixel-based analysis of elastographic maps.”
We have fixed the missing parenthesis in the ROC curve statement. The corrected sentence now reads:
"The Receiver Operating Characteristic (ROC) curve model also indicated a higher predictive value."
#Point 2. : 1. Introduction
“...comprising more than 20% of vaginal deliveries[1].” – The percentage varies significantly depending on the country studied, as noted in WHO data. I recommend adding this context.
Response: We agree that the IOL rate varies significantly across different countries and healthcare systems. We will revise the sentence to reflect this global variation:
"Induction of labor (IOL) is stimulation of contraction before the spontaneous onset of labor and is one of the most common procedures in obstetrics, with rates varying significantly across countries, ranging from 20% to 35% of deliveries according to previous data[1-3]."
#Point 3. “Rates in the United States have increased significantly from 9.6% in 1990 to 25.7% in 2018.” – It would be worthwhile to reference more recent data than 2018, as these figures are updated annually in U.S. statistical reports.
Response: Thank you for your suggestion to include more recent data on labor induction rates in the United States. Upon reviewing the latest available statistics, I have updated the manuscript accordingly.
Revised manuscript:
“Labor induction rates in the United States have increased significantly from 9.6% in 1990 to 4125.7% in 202218 reflecting a continued upward trend in obstetric interventions.[4, 5]”
#Point 4. Phrases such as “ruptured membranes without labor” and “pregnancies that have gone past their due date” should use standardized medical terminology, such as Postterm pregnancy or PROM (Premature Rupture of Membranes).
Response: Thank you for your feedback regarding the medical terminology. We agree that using standardized medical terminology enhances the scientific rigor and clarity of the manuscript. We have revised these phrases as follows:
"Ruptured membranes without labor" has been changed to "Premature Rupture of Membranes (PROM)"
"Pregnancies that have gone past their due date" has been changed to "postterm pregnancy"
#Point 5. The Bishop score has been the sole method used to evaluate cervical status; however, it is subjective and causes discomfort to patients.” – In my opinion, the main limitation of the Bishop score is its lack of reproducibility in quantitative terms and the significant variability between examiners (see Faltin-Traub, E.F.; Boulvain, M.; Faltin, D.L.; Extermann, P.; Irion, O. Reliability of the Bishop score before labour induction at term. Eur. J. Obstet. Gynecol. Reprod. Biol. 2004, 112, 178–181). There is ongoing research aimed at objectifying this scale by incorporating it into ultrasound frameworks, including the use of E-Cervix software (DOI: 10.3390/jcm12134492). I believe this is a point worth addressing in the introduction.
Response: Thank you for highlighting this important point. You are absolutely right that we should emphasize the reproducibility and inter-examiner variability issues of the Bishop score, rather than focusing solely on subjectivity and patient discomfort. Here is my revision of that section:
To date, the Bishop score has been the traditional method used to evaluate cervical status; however, it is subjective and causes discomfort to patients[11] and its primary limitation lies in its poor reproducibility and significant inter-examiner variability.[12] Recent research has focused on incorporating the Bishop score into ultrasound frameworks, including the development of E-Cervix software, to achieve more objective measurements.[13] Despite these advances, there remains a need for more objective methods to predict cervical characteristics.
#Point 6. In the introduction, the authors also discuss SWE (shear wave elastography). It would be helpful to differentiate between SWE and strain elastography, as this distinction is key to understanding the E-Cervix module.
Response: Thank you for this valuable suggestion. You're right that we should clearly differentiate between SWE and strain elastography. I have updated the manuscript accordingly.
Revised manuscript: Along with cervical length, several elastography tools have been developed for the objective assessment of cervical condition. Two main elastography techniques are currently used: strain elastography and shear wave elastography (SWE). While strain elastography measures tissue deformation in response to external compression and provides qualitative or semi-quantitative assessments, SWE uses acoustic radiation force to generate shear waves and provides quantitative measurements of tissue stiffness in kilopascals (kPa).[15] SWE is an ultrasound-based method that allows for direct estimation of cervical tissue density without the need for manual compression.[16, 17]
#Point 7. Methods In this section, the authors should first describe the protocol for labor induction. Specifically, detail the criteria for oxytocin infusion initiation (e.g., based on the Bishop score), whether preinduction methods were used (e.g., prostaglandins, laminaria, Foley catheter), and under what circumstances amniotomy was performed.
Response: Thank you for your valuable feedback regarding the labor induction protocol description. We agree that a more detailed explanation of our induction protocol would enhance the reproducibility and clarity of our study. We have revised the methods section to include comprehensive information about our institutional protocol for labor induction.
2.2 Protocol for labor induction
Labor induction was performed according to our institutional protocol. Prior to induction, all patients underwent cervical assessment using the Bishop score. For patients with an unfavorable cervix (Bishop score ≤ 6), cervical ripening was initiated using either prostaglandin E2 (dinoprostone vaginal insert, 10 mg). Oxytocin infusion was initiated either after cervical ripening or directly in patients with a favorable cervix (Bishop score > 6). Oxytocin was administered using a standardized protocol starting at 2 mU/min, with increments of 2 mU/min every 30 minutes until adequate uterine contractions were achieved (3-5 contractions per 10 minutes) or a maximum dose of 20 mU/min was reached. Artificial rupture of membranes (amniotomy) was sometimes performed when the cervix was favorable (≥ 4 cm dilated) and the fetal head was engaged to the cervix, unless spontaneous rupture had already occurred. Throughout the induction process, continuous fetal heart rate monitoring was maintained, and the progress of labor was assessed by cervical examination every hour or as clinically indicated.
#Point 8. “The detailed descriptions of each parameter are provided in a separate paper.” – A reference to the appropriate publication must be provided. For instance, the parameters are described here: (DOI: 10.1038/s41598-021-02498-3).
Response: Thank you for pointing out the need for proper referencing of the parameter descriptions. We agree that providing a specific reference is crucial for readers to access detailed parameter information. We have added the appropriate citation in the manuscript.
#Point 9. The methodology should also include a more detailed description of the imaging technique used for E-Cervix evaluation. In this context, the pressure applied by the probe to the cervix is critically important and should be specified.
Response: We appreciate this important point about the imaging technique details, particularly regarding probe pressure. We have added comprehensive information about our standardized imaging protocol, including specific details about probe pressure control and measurement techniques.
Here is my revision of that section:
Our standardized protocol for E-cervix elastography measurements included several key steps to ensure optimal image acquisition and quality:
- Pre-examination Preparation
Patients were instructed to void their bladder before the examination to optimize image quality. All examinations were performed with the patient in the lithotomy position.
- Image Acquisition Technique
The ultrasound probe was positioned to display the cervical apex at the monitor's superior aspect, with fetal components oriented to the left of the image sector. Following Fetal Medicine Foundation guidelines, we obtained cervical images using the same plane as standard cervical length measurement, taking care to minimize anterior cervical pressure.
- Quality Assurance
Image quality was monitored through the system's reliability indicators. The probe was maintained in a stable position until all motion indicators displayed acceptable status (green indicators). Patients were instructed to maintain normal breathing patterns during image acquisition. Images affected by fetal movement, particularly in cases of breech presentation, were discarded and repeated to ensure measurement accuracy.
- Region of interest (ROI) Placement Protocol
ROI markers were positioned using the grayscale reference image for optimal precision.
The endocervical canal was delineated using either a two-point or four-point ROI system, depending on cervical curvature. The measurement area was carefully defined to encompass the entire cervical tissue while excluding adjacent structures such as bladder tissue or vaginal wall.
#Point 10. Results
“Among them, 29 reached the active phase, while 42 failed to enter the active phase.” – Percentages should be provided here.
Response: Thank you for pointing out the need to include all percentages. Here is the corrected version:
"After excluding ineligible patients, 71 women were included in the study. Among them, 29 (40.8%) reached the active phase, while 42 (59.2%) failed to enter the active phase."
#Point 11. The term posterior cervical angle appears in the tables for the first time but is not mentioned earlier in the text. This term should be introduced and explained earlier in the manuscript.
Response: Thank you for highlighting this oversight regarding the posterior cervical angle. You are correct that this parameter should be properly introduced and explained before appearing in the tables.
“In addition to cervical length, we measured the posterior cervical angle, which is defined as the angle between the cervical canal and the posterior uterine wall.”
#Point 12. Given the low reproducibility of the Bishop score between observers, it would be helpful to present it as a categorical variable in the results. For instance, literature commonly uses a cutoff of BS <7 and ≥7 to define an unfavorable or favorable cervix, respectively.
Response: Thank you for this thoughtful suggestion regarding the Bishop score categorization. However, we need to clarify an important characteristic of our study population.
In our study, all participants had a Bishop score < 7, as this was one of our inclusion criteria for patient selection. Therefore, while we appreciate the value of categorizing the Bishop score using the traditional cutoff of 7 to distinguish between favorable and unfavorable cervix, this categorization would not be applicable to our analysis since our entire study population fell into the "unfavorable cervix" category. This homogeneity in terms of Bishop scores was intentional in our study design to specifically evaluate prediction models in cases where the cervix was unfavorable.
This is precisely why we focused on identifying other potential predictive factors and developing more objective measurements for this specific population, where traditional Bishop score categorization may have limited discriminative value.
#Point 13. Discussion
“Previous studies have shown that elastography can be a useful tool for predicting the onset of labor induction or the success of vaginal delivery.” – As mentioned earlier, attempts have been made to reflect the Bishop score using ultrasound parameters and the E-Cervix module. This effort to objectify the evaluation before induction of labor (IOL) should be mentioned in the discussion (DOI: 10.3390/jcm12134492).
Response: We appreciate your suggestion to include discussion of recent efforts to objectify pre-induction evaluation using ultrasound parameters and the E-Cervix module. We have expanded our discussion to reflect these important developments.
Revised manuscript:
Previous studies have shown that elastography can be a useful tool for predicting the onset of labor induction or the success of vaginal delivery. Lu et al.[10] concluded that the combination of sonographic CL and SWE is superior to the Bishop score in predicting IOL failure. Recent advances have focused on integrating ultrasound parameters with traditional assessment methods to create more objective evaluation systems. In this point of view, oOur study enhances previous findings in different ways. First, we created models to illustrate how the AUC changed when the sonographic and elastographic parameters were added. Second, the DeLong test was used to evaluate the significance of each model.
E-cervix(Samsung Medison Co., Ltd.,) is a semi-automatic software that analyzes the strain ratio between the internal and external orifices of the cervix using vibrations caused by natural internal movements. The advantage of this technique is that the generation of the mechanical impulse is operator independent, which improves reproducibility and reduces interobserver variability[22]. Using E-cervix, which measures cervical stiffness, others have suggested its ability to predict spontaneous preterm delivery[23], success of IOL[11], and failure to enter the active phase[10]. This module combines various ultrasound parameters to provide a more reproducible and objective evaluation of cervical status, addressing the limitations of the traditional Bishop score. The effort to translate subjective clinical parameters into quantifiable ultrasound measurements has shown promising results in predicting induction outcomes.Based on prior research, we compared the predictive performance curves based on maternal baseline characteristics with sonographic and elastographic parameters. Our analysis demonstrated higher performance values for elastographic parameters.
#Point 14. The authors should provide a broader literature context for their study. For example, a Medline search using the keyword "E-Cervix" yields 27 results.
Response: Thank you for suggesting a more comprehensive literature review. We agree that providing broader context would strengthen our manuscript. We have conducted a thorough literature review of E-Cervix-related research and would like to address this in our discussion. We have revised our discussion to incorporate these findings and place our research within this broader context.
E-cervix(Samsung Medison Co., Ltd.,) is a semi-automatic software that analyzes the strain ratio between the internal and external orifices of the cervix using vibrations caused by natural internal movements. The advantage of this technique is that the generation of the mechanical impulse is operator independent, which improves reproducibility and reduces interobserver variability[22]. A comprehensive literature review of E-Cervix-related research reveals several key developments in this field. Early studies focused on validating the E-Cervix module as an objective tool for cervical assessment, demonstrating its potential to overcome the limitations of traditional Bishop score evaluation.[19, 23-25] Several studies have explored the predictive value of E-Cervix measurements for various obstetric outcomes. Particularly noteworthy is the emerging evidence regarding the hardness ratio (HR) as a predictive marker. Nazzaro et al. (2024) demonstrated that women with low HR, especially those with values less than 50% or 35%, showed an increased risk of preterm birth (PTB). Their findings revealed that women who delivered preterm had significantly higher HR compared to those who carried to term, along with notably lower internal os strain (IOS) and external os strain (EOS).[17, 26] The predictive value of elastography extends beyond just preterm birth prediction. Rizzo et al. found that HR assessment through sonoelastography enhances the predictive accuracy of cervical length measurements for imminent delivery in nulliparous women approaching term. This suggests that combining traditional cervical length measurements with elastography parameters might provide more comprehensive risk assessment.[22] Furthermore, He et al. emphasized that in singleton pregnancies with a short cervix receiving progesterone therapy, cervical length measurement alone may be insufficient for accurate risk assessment of spontaneous preterm birth.[27] Their research highlighted the importance of evaluating cervical stiffness, particularly focusing on internal and external os strain measurements. In specific high-risk populations, such as women with a history of Loop Electrosurgical Excision Procedure (LEEP), Cha et al. demonstrated that cervical strain measured in mid-trimester could be particularly valuable. They found that previous LEEP procedures were associated with changes in cervical strain and cervical length shortening, establishing elastography as a useful tool for predicting sPTB in this population.[28] Furthermore, others have suggested its ability to predict spontaneous preterm delivery[29], success of IOL[11], and failure to enter the active phase[10]. This module combines various ultrasound parameters to provide a more reproducible and objective evaluation of cervical status, addressing the limitations of the traditional Bishop score. The effort to translate subjective clinical parameters into quantifiable ultrasound measurements has shown promising results in predicting induction outcomes. Based on prior research, we compared the predictive performance curves based on maternal baseline characteristics with sonographic and elastographic parameters. Our analysis demonstrated higher performance values for elastographic parameters.
#Point 15. Additionally, the discussion should address the limitations of the study. In my opinion, one limitation might be the reproducibility of E-Cervix results. While the parameters generally show good reproducibility, the poorest reproducibility is observed for EOS, which affects the reliability of the IOS/EOS parameter included in the multivariable model (DOI: 10.3390/jcm12134492).
Response:
We agree that addressing the reproducibility of E-Cervix measurements, particularly EOS measurements, is crucial for a comprehensive discussion of our study limitations. We have added a detailed paragraph addressing this limitation.
Revised manuscript:
Third, a key limitation concerns the reproducibility of E-Cervix measurements, particularly the EOS parameter. As demonstrated by Mlodawski et al., while most E-Cervix parameters show good reproducibility, EOS measurements demonstrate relatively lower reproducibility compared to other parameters.[13] This variability in EOS measurements could potentially affect the reliability of the IOS/EOS ratio used in our multivariable model. To minimize this limitation in our study, all measurements were performed by experienced sonographers who underwent standardized training, and multiple measurements were taken for each parameter. However, future studies should consider this inherent variability when interpreting EOS-related measurements and perhaps explore alternative parameters or measurement techniques that might offer better reproducibility.
#Point 16. General Comments: Overall, the study is good and relatively innovative, as there is limited literature on this method. However, it requires some improvements.
Response: Thank you for your thoughtful and encouraging evaluation of our study. We appreciate your recognition of our work's innovative nature and contribution to the limited existing literature on E-Cervix methodology in labor induction prediction.
We have carefully addressed all your suggested improvements throughout the manuscript, including: Adding more recent data for labor induction rates, implementing standardized medical terminology, providing detailed descriptions of our labor induction protocol, including appropriate references for parameter descriptions, adding comprehensive details about the E-Cervix imaging technique, expanding the literature context in our discussion, acknowledging study limitations, particularly regarding EOS measurement reproducibility.
These revisions have significantly strengthened the manuscript while maintaining its innovative contribution to the field. We believe these changes have enhanced the clarity, reproducibility, and scientific rigor of our work.
Reviewer 2 Report
Comments and Suggestions for Authors
Dear Authors,
I want to thank you for sharing this exciting work and to applaud your efforts to elaborate on this paper.
In my opinion, the article needs some supplementary work to meet the journal‘s standards.
- Abstract: ok.
- Keywords: ok
- Introduction: In the introduction, you could elaborate on the topic of labor induction and the conditions for its initiation.
- Methods and results:
* In the inclusion criteria, the age of the patients should be specified.
* It is essential to mention the reason for labor induction, the method used to achieve it, the doses, the administration time, and the period or duration until patients enter the active phase of labor.
* Table 1 is not mentioned in the text, and the date included in the table is not described.
* Figure number 1 does not appear in the document. Just so you know – only the description of the figure is provided, but the figure itself is not attached.
- Discussion: The paragraph containing the discussion is very brief and does not provide enough information about the topic addressed in this article. Please elaborate on the subject and attach the appropriate bibliography, as I believe the number of references is insufficient.
- Conclusion: ok.
- Reference: More references should be cited for such an important topic.
Best regards.
Author Response
Reviewers' Comments 2
#Point 1. I want to thank you for sharing this exciting work and to applaud your efforts to elaborate on this paper. In my opinion, the article needs some supplementary work to meet the journal‘s standards.
- Abstract: ok.
- Keywords: ok
- Introduction: In the introduction, you could elaborate on the topic of labor induction and the conditions for its initiation.
Response: Thank you for suggesting the need for more context regarding labor induction in our introduction. We agree that providing more detailed information about labor induction indications and initiation criteria would strengthen our manuscript.
We have expanded our introduction to include: 1) Common medical indications for labor induction (such as postterm pregnancy, premature rupture of membranes, gestational hypertension, and other maternal-fetal conditions), 2) The importance of appropriate patient selection for labor induction, 3) The current criteria used to determine timing of induction initiation, 4) The relationship between successful induction and initial cervical status
#Point 2. Methods and results:
* In the inclusion criteria, the age of the patients should be specified.
Response: We agree that specifying the age range of eligible patients is important for clarity and reproducibility. We have added this information to our inclusion criteria.
Revised manuscript:
Inclusion criteria were as follows: 1) singleton pregnant women aged 19-45 years; 2) gestational age ≥37 weeks; 3) women with no contraindications for IOL; and 4) women with unfavorable cervix (Bishop score < 7).
#Point 3. It is essential to mention the reason for labor induction, the method used to achieve it, the doses, the administration time, and the period or duration until patients enter the active phase of labor.
Response: We appreciate your valuable comment regarding the need for comprehensive information about labor induction. We have made the following additions to our manuscript:
In the Methods section, we have detailed our institutional labor induction protocol, including:
1) The specific methods used for cervical ripening; 2) Oxytocin administration protocol; 3) Criteria for amniotomy, and 4) Monitoring procedures
In the Results section, we have added a detailed labor induction indications in our study population. Regarding the duration until active phase, the median time to active phase was 10 hours (IQR 8-14).
Revised manuscript:
2.2 Protocol for labor induction
Labor induction was performed according to our institutional protocol. Prior to induction, all patients underwent cervical assessment using the Bishop score. For patients with an unfavorable cervix (Bishop score ≤ 6), cervical ripening was initiated using either prostaglandin E2 (dinoprostone vaginal insert, 10 mg). Oxytocin infusion was initiated either after cervical ripening or directly in patients with a favorable cervix (Bishop score > 6). Oxytocin was administered using a standardized protocol starting at 2 mU/min, with increments of 2 mU/min every 30 minutes until adequate uterine contractions were achieved (3-5 contractions per 10 minutes) or a maximum dose of 20 mU/min was reached. Artificial rupture of membranes (amniotomy) was sometimes performed when the cervix was favorable (≥ 6 cm dilated) and the fetal head was engaged to the cervix, unless spontaneous rupture had already occurred. Throughout the induction process, continuous fetal heart rate monitoring was maintained, and the progress of labor was assessed by cervical examination every hour or as clinically indicated. For patients who failed to enter active labor after the initial induction, the same induction protocol was repeated 24 hours later.
3.1. Study population
After excluding ineligible patients, 71 women were included in the study. The indications for labor induction in our study population included premature rupture of membranes (PROM) (n=20, 28.2%), low-risk nulliparous women after 39 weeks of gestation (n=27, 38.0%), gestational diabetes mellitus (GDM) (n=6, 8.5%), hypertensive disorders of pregnancy (HDP) (n=5, 7.0%), intrauterine growth restriction (IUGR) (n=6, 8.5%), large for gestational age (LGA) (n=2, 2.8%), and postterm pregnancy (n=5, 7.0%). Among them, 29 (40.8%) reached the active phase, while 42 (59.2%) failed to enter the active phase. In the group that entered the active phase, 86.2% successfully achieved a vaginal delivery. The median time to active phase was 10 hours (IQR 8-14).
#Point 4. Table 1 is not mentioned in the text, and the date included in the table is not described.
Response: We have addressed the concern about Table 1 not being referenced in the text by incorporating mentions of Table 1 in our Results section where we present the study population characteristics and induction indications.
#Point 5. Figure number 1 does not appear in the document. Just so you know – only the description of the figure is provided, but the figure itself is not attached.
Response: Thank you for bringing this to our attention regarding Figure 1. There appears to have been a technical issue during the submission process where the figure file was not properly attached, although we did include it in our submission. We confirm that the figure was prepared and will ensure it is properly attached in this revision.
#Point 6. Discussion: The paragraph containing the discussion is very brief and does not provide enough information about the topic addressed in this article. Please elaborate on the subject and attach the appropriate bibliography, as I believe the number of references is insufficient.
Response: Thank you for highlighting the need for a more comprehensive discussion section. We have significantly expanded our discussion to provide a more thorough analysis of our findings in the context of current literature:
We have added an extensive review of E-Cervix-related research, incorporating findings from the published studies identified through our Medline search.We have expanded our discussion of cervical elastography's role in various clinical contexts, including: The relationship between hardness ratio (HR) and preterm birth risk, The enhanced predictive value of combining elastography with cervical length measurements, he importance of cervical stiffness evaluation in progesterone-treated patients, the role of elastography in patients with previous LEEP procedures.
We have included a detailed discussion of study limitations, particularly addressing the reproducibility of E-Cervix measurements, especially regarding EOS parameters. We have strengthened our bibliography by adding relevant recent publications and key foundational studies in the field. These additions provide a more comprehensive analysis of our findings within the broader context of current research in cervical assessment and labor induction prediction.
#Point 7. Reference: More references should be cited for such an important topic.
Response: Thank you for mentioning the need for more comprehensive referencing in our manuscript. We agree that this important topic warrants additional literature support. We have significantly expanded our reference list to include:
1) Recent key studies on E-Cervix technology and its applications; 2) Foundational research on cervical elastography; 3) Current literature on labor induction outcomes; and 4) Studies addressing the limitations and reliability of traditional cervical assessment methods
Round 2
Reviewer 2 Report
Comments and Suggestions for Authors
Dear Authors, your consistent efforts to improve the manuscript in my opinion made it eligible for publication.
Author Response
#Point 1. Dear Authors, your consistent efforts to improve the manuscript in my opinion made it eligible for publication.
Response: Thank you for your positive evaluation. We greatly appreciate your recognition of our efforts to improve the manuscript. Your feedback throughout the review process has been invaluable in helping us enhance the quality of our work. We are pleased that the revisions have met the publication standards.